# A Comparative Evaluation of Desoximetasone Cream and Ointment Formulations Using Experiments and In Silico Modeling

**DOI:** 10.3390/ijms242015118

**Published:** 2023-10-12

**Authors:** Namrata S. Matharoo, Harsha T. Garimella, Carrie German, Andrzej J. Przekwas, Bozena Michniak-Kohn

**Affiliations:** 1Center for Dermal Research, Rutgers, The State University of New Jersey, Piscataway, NJ 08854, USA; nsm112@scarletmail.rutgers.edu; 2Ernest Mario School of Pharmacy, Rutgers, The State University of New Jersey, Piscataway, NJ 08854, USA; 3CFD Research Corporation, Huntsville, AL 35806, USA

**Keywords:** topical, transdermal, formulation, drug release, permeation, in silico modeling, IVRT, IVPT, PBPK, PKPD, modelling and simulation, pharmacology, dermal kinetics

## Abstract

The administration of therapeutic drugs through dermal routes, such as creams and ointments, has emerged as an increasingly popular alternative to traditional delivery methods, such as tablets and injections. In the context of drug development, it is crucial to identify the optimal doses and delivery routes that ensure successful outcomes. Physiologically based pharmacokinetic (PBPK) models have been proposed to simulate drug delivery and optimize drug formulations, but the calibration of these models is challenging due to the multitude of variables involved and limited experimental data. One significant research gap that this article addresses is the need for more efficient and accurate methods for calibrating PBPK models for dermal drug delivery. This manuscript presents a novel approach and an integrated dermal drug delivery model to address this gap that leverages virtual in vitro release (IVRT) and permeation (IVPT) testing data to optimize mechanistic models. The proposed approach was demonstrated through a study involving Desoximetasone cream and ointment formulations, where the release kinetics and permeation profiles of Desoximetasone were determined experimentally, and a computational model was created to simulate the results. The experimental studies showed that, even though the cumulative permeation of Desoximetasone at the end of the permeation study was comparable, there was a significant difference seen in the lag time in the permeation of Desoximetasone between the cream and ointment. Additionally, there was a significant difference seen in the amount of Desoximetasone permeated through human cadaver skin at early time points when the cream and ointment were compared. The computational model was optimized and validated, suggesting that this approach has the potential to bridge the existing research gap by improving the accuracy and efficiency of drug development processes. The model results show a good fit between the experimental data and model predictions. During the model optimization process, it became evident that there was variability in both the permeability and the partition coefficient within the stratum corneum. This variability had a significant and noteworthy influence on the overall performance of the model, especially when it came to its capacity to differentiate between cream and ointment formulations. Leveraging virtual models significantly aids the comprehension of drug release and permeation, mitigating the demanding data requirements. The use of virtual IVRT and IVPT data can accelerate the calibration of PBPK models, streamline the selection of the appropriate doses, and optimize drug delivery. Moreover, this novel approach could potentially reduce the time and resources involved in drug development, thus making it more cost-effective and efficient.

## 1. Introduction

The administration of therapeutic drugs through the dermal route has emerged as a potential substitute for traditional delivery routes, including oral tablets and parenteral injections such as intravenous or intramuscular routes [1,2]. Among the available transdermal products in the market, passive delivery systems, such as patches, gels, creams, and sprays, are the most commonly used [3]. The efficacy of dermal delivery and subsequent transdermal delivery of drugs is primarily dependent on the characteristics of the delivery vehicle and the resistance offered by the skin [1]. Experimental studies can provide valuable information about drug delivery, but they can be expensive, time-consuming, and may not always be feasible for some drug candidates. It is essential to explore new methods that can provide accurate predictions for drug delivery without incurring high costs. Such alternative methods could be particularly useful in evaluating the efficacy of drug delivery systems in the preclinical stage, which could minimize the potential risks associated with clinical trials [4,5]. Hence, there is a need for cost-effective modeling approaches that can simulate the transport of drugs through the skin and predict the performance of transdermal drug delivery systems.

Physiologically based pharmacokinetic (PBPK) models have emerged as a cost-effective alternative to experimental approaches in drug development. These models have become increasingly sophisticated and are being employed to inform the selection of the appropriate doses, optimize drug delivery, and make other necessary adjustments. Moreover, the US Food and Drug Administration (FDA) has acknowledged the vital role that PBPK models can play in supporting the development of alternative bioequivalence (BE) approaches and drug approval. However, while PBPK models have shown great potential in predicting drug exposure and providing valuable insights into pharmacokinetic and pharmacodynamic properties, their accuracy is still limited by the quality and availability of the input data used for the model optimization. Therefore, there is a need to develop an integrated experimental computational approach that combines experimental data with computational modeling to improve the accuracy and predictive power of PBPK models. This integrated approach could provide more reliable predictions of drug exposure and efficacy, which could reduce the time and costs involved in drug development, as well as the risks associated with clinical trials.

Prior research in the field of dermal drug delivery has laid the foundation for the current study’s objectives and research gap. Numerous research groups have developed mathematical models to understand drug transport within specific skin layers, including the stratum corneum (SC), viable epidermis (VE), and dermis (DM) [6,7,8,9,10,11,12,13,14,15]. Mechanistic models are a type of computational model that uses mathematical descriptions of the underlying biological and physicochemical processes involved in drug delivery. These models have contributed significantly to our knowledge, but have often existed independently. Very few attempts have been made to integrate them into a comprehensive skin model capable of reliably quantifying drug distribution across all skin layers and into the systemic circulation. Previous models have sometimes oversimplified the skin’s complexity or lacked practical utility. While some models have explored dermal absorption and systemic pharmacokinetics, they have not considered the full range of organs and anatomical structures.

In this direction, in our previous work [3], we developed a comprehensive mechanistic skin model coupled with dermal drug delivery system (D3S models), which was subsequently linked to a whole-body physiologically based pharmacokinetic (PBPK) model [3]. Our integrated model addresses these limitations by linking a holistic skin model (SC + VE + DM) with a systems pharmacology model. This model enables precise predictions of dermal absorption, clinical pharmacokinetics, and the combined effects of concomitant medications due to patient comorbidities. More information about this model, including its advantages and limitations, can be obtained from our work [3].

In the current study, we utilized the integrated dermal drug delivery mechanistic model developed in our previous work [3], along with release and permeation experimental data, corresponding to Desoximetasone, to address several central questions in the field of dermal drug delivery, including: (a) Different Formulation Impacts: How do cream and ointment formulations impact the process of dermal drug delivery, including factors such as absorption, permeation through various skin layers, and the optimization of drug formulations and delivery routes? (b) Simulation of Formulations: Can mechanistic models effectively simulate the distinct characteristics of cream and ointment formulations in dermal drug delivery, considering their unique release kinetics and permeation profiles?

Numerous studies have examined the efficacy of IVRT and IVPT experiments in assessing the permeation and release characteristics of pharmaceutical creams, ointments, gels, patches, and others [16,17,18,19]. While IVPT primarily focuses on evaluating the skin penetration and absorption of active compounds, IVRT concentrates on understanding the drug release behavior of the formulation. While prior research has made significant strides in elucidating the impacts of different formulations on dermal drug delivery, there are still some gaps in knowledge that persist. Notably, previous studies have often lacked a comprehensive integration of IVRT and IVPT experimental data (IVRT and IVPT) with mechanistic modeling (modeling of both the skin and the formulations). Pensado et al. [19] used a simplified skin model when using both IVRT/IVPT data.

To enable the practical application of this mechanistic integrated model for in vivo simulations, rigorous optimization and validation in vitro are necessary. However, calibrating mechanistic models can be challenging due to the large number of variables involved and the limited experimental data that are typically available for optimization. The accuracy of these models heavily depends on the available knowledge about the morphological and physicochemical properties of the skin. Therefore, to increase the scope and fidelity of such models, it is important to calibrate them more rigorously by incorporating experimental data. In this article, we propose a novel approach to optimizing or calibrating mechanistic models for transdermal drug delivery using a splitting approach that calibrates the dosage and skin models separately using in vitro release (IVRT) and permeation (IVPT) testing data, respectively. This approach allows us to focus on optimizing specific parameters in each part, rather than trying to optimize all the parameters simultaneously, making the optimization process more efficient and effective.

Our research focuses on addressing these pivotal questions to advance our understanding of dermal drug delivery processes and improve the utility of PBPK models for predicting transdermal drug delivery. The model developed here has the potential to significantly enhance the accuracy and efficiency of drug development processes, while also mitigating the risks associated with clinical trials. Additionally, our framework serves as a valuable tool for future research endeavors aimed at enhancing the fidelity of virtual drug delivery models in this critical field.

In this paper, a schematic of the model optimization and validation process using in vitro experimental data is presented in Figure 1. The model optimization process was performed by first using in vitro release testing (IVRT) data to calibrate and validate the mechanistic release model parameters. The calibrated parameters were then implemented into the virtual in vitro permeation testing (IVPT) model for the further optimization and validation of the dermal model. The IVRT and IVPT experimental data used in this study were generated at the Center for Dermal Research at Rutgers University. The computational modeling and validation were performed by the CFD Research team. Section 2 of the paper introduces the overall experimental and modeling approach, including the different experimental protocols, components of the formulation model, components of the integrated dermal absorption model, and governing equations. Additional details about the dermal physiologically based pharmacokinetic (PBPK) model can be found in Somayaji et al. [3]. Section 3 of the paper includes the experimental data and model predictions of the drug concentration in different compartments. The results of the study are discussed in Section 4, including the implications of the study’s findings. The experiments revealed notable differences between the cream and ointment formulations of Desoximetasone, particularly in terms of lag time and early-stage permeation through human cadaver skin. The computational model was effectively validated, indicating its potential to enhance drug development accuracy and efficiency. The key findings highlighted distinct release and permeation profiles influenced by excipients. The lag times correlated with the formulation properties, and Desoximetasone exhibited a preference for epidermal deposition. Utilizing virtual models facilitates understanding of drug release and permeation, simplifying the data requirements. Virtual IVRT and IVPT data can expedite PBPK model calibration, dose selection, and delivery optimization. This innovative approach holds promise for streamlining drug development, potentially reducing costs and enhancing efficiency. Section 4 also discusses the limitations of the study in detail, and proposes future research directions to address these limitations. In summary, while the computational model in this study advances our understanding of cream and ointment formulations’ drug release, it has limitations. It does not consider formulation viscosity, which impacts release and penetration. Incorporating viscosity would require additional experimental data. The model does not include particle size effects, pending further research. It also lacks data on formulation composition, which could enhance accuracy if available. Future versions may address these limitations for improved reliability. The paper concludes with Section 5, which summarizes the study’s main findings and contributions to the field of drug delivery modeling. A list of abbreviations used in the manuscript is included below for reference.

## 2. Results

### 2.1. IVRT: Experimental Results

The study performed in vitro release testing using Desoximetasone cream, ointment, and 0.25% Desoximetasone. These formulations/dilutions were applied to an inert SnakeSkin Dialysis tubing membrane. The results (Figure 2) indicated that the cream formulation exhibited a higher release rate compared to the ointment formulation. However, the release profiles did no indicate a significant difference (*p* > 0.05), which may indicate that the bioavailability of the Desoximetasone from the two formulations was similar over the tested time frame. The observed difference in the release kinetics could be attributed to the differences in the physicochemical properties of the excipients used in the formulations, which can impact the solubility and diffusivity of the active ingredient. Additionally, the results suggested that the behavior of the pure solute was affected by the other excipients in the formulations. This indicates that the excipients used in the formulations had a significant impact (*p* < 0.05) on the release kinetics of the Desoximetasone.

### 2.2. IVPT: Experimental Results

#### 2.2.1. Desoximetasone Permeation Studies

Ex vivo skin permeation studies were carried out with both Desoximetasone cream and ointment. These formulations were applied to human cadaver skin using Franz Diffusion Cells (FDC). These techniques yielded plots for cumulative drug permeation versus time, steady flux state, and permeability coefficients (Figure 3). Similar permeation profiles were observed with both the cream and the ointment. A statistical analysis revealed that the amount of Desoximetasone permeated per cm^2^ of skin was significantly higher in the cream as compared to the ointment from the time point of 12 h to 18 h (*p* < 0.05) (Table 1). Additionally, it was also observed that the lag time of the permeation of the Desoximetasone through the human cadaver skin was shorter for the cream as compared to the ointment. This may indicate that Desoximetasone may be bioavailable faster in cream as compared to ointment.

#### 2.2.2. Desoximetasone Absorption in Skin

Post-IVPT, the skin exposed to the applied dose was collected and separated into the dermis and epidermis using forceps. The separated layers were weighed prior to any treatment for mass balance. The Desoximetasone was extracted and quantified from the skin layers using 100% methanol (Section 2.2.3). This technique yielded a comparative view of the Desoximetasone deposition in the epidermis and dermis, which is presented in Figure 4. The amount of Desoximetasone in the epidermis was significantly higher than that in the dermis with both the cream and the ointment. (*p* < 0.05) However, the amount of Desoximetasone from the cream vs. ointment in the epidermis and dermis showed no significance. (*p* > 0.05).

The recovery was calculated with respect to the initial amount of Desoximetasone from the same weight of formulation. The amount of Desoximetasone was found to be similar in both formulations and the recovery of Desoximetasone from the cream ranged from 90 to 97% (*n* = 5) and from the ointment it ranged from 97 to 100% (*n* = 5). The differences in recovery can be attributed to some interferences from skin components which may have degraded or diffused into the receptor compartment over the time.

#### 2.2.3. IVRT: Experiments vs. Simulations

After the development of the virtual in vitro release testing (IVRT) computational models, the model prediction of the amount of Desoximetasone permeated into the receptor compartment was compared against the experimental data for both the cream and ointment formulations. As described in the Methods section, the model parameters were optimized during the comparison process, ensuring that the model predictions agreed with the experimental data. The results from the virtual IVRT models (shown in Figure 5) demonstrated that the optimized release model parameters were able to accurately reproduce the experimentally observed release behavior with minimal optimization. Figure 5A shows the virtual IVRT model predictions compared against the experimental data for the Desoximetasone ointment (0.25% Actavis). Similarly, Figure 5B displays the same for the Desoximetasone 0.25% cream (Perrigo). At the different time points, there was a drop observed in the concentration of the receptor. This was mainly due to the effect of sampling from the receiver compartment. The optimized release model parameters were subsequently employed in the virtual in vitro permeation testing (IVPT) models to simulate the permeation of Desoximetasone across the skin. It is worth noting that the use of IVRT to optimize the release model parameters enabled us to constrain our model optimization during the IVPT validation process. The findings from this study underscore the potential of in silico modeling as a means of streamlining the drug development process, by reducing the need for extensive experimental testing while ensuring the accuracy and reliability of predictions. Future research in this area could build upon the current study by incorporating additional factors that may impact drug release and permeation, such as co-solvent effects and the impact of the formulation parameters.

#### 2.2.4. IVPT: Experiments vs. Simulations

Figure 6 and Figure 7 show the model predictions for the amount permeated into the receptor and the drug accumulation in the different skin layers compared against the experimental data for the Actavis cream and Perrigo ointment, respectively. The drug accumulated included the drug accumulated in the epidermis and dermis layers. The release model used the fine-tuned release model parameters from the above section. Upon the minimal optimization of the skin model, it appeared that the model predictions compared well against the experimental data.

Based on the data, for the cream model, the amount permeated into the receptor compared well with the experimental data (Figure 6A). The results of the in silico simulation and experimental data were also in close agreement for both the epidermis and dermis layers of the skin. Specifically, the simulated value for the amount of Desoximetasone permeated into the epidermis layer (Figure 6B) was 5.09 × 10^−1^ (µg/mg), which is comparable to the experimental value of 5.00 × 10^−1^ (µg/mg). Similarly, the simulated and experimental values for the amount of Desoximetasone permeated into the dermis layer were 1.79 × 10^−2^ (µg/mg) and 2.00 × 10^−2^ (µg/mg), respectively, which are also in close agreement (Figure 6B).

Based on the data, for the ointment model, the amount permeated into the receptor compared well with the experimental data (Figure 7A). The results of the in silico simulation and experimental data were also in close agreement for both the epidermis and dermis layers of the skin. Specifically, the simulated value for the amount of Desoximetasone permeated into the epidermis layer (Figure 7B) was 6.97 × 10^−1^ (µg/mg), which is comparable to the experimental value of 7.00 × 10^−01^ (µg/mg). Similarly, the simulated and experimental values for the amount of Desoximetasone permeated into the dermis layer were 2.06 × 10^−2^ (µg/mg) and 2.00 × 10^−2^ (µg/mg), respectively, which are also in close agreement (Figure 7B).

Table 2 presents the mechanistic estimation and calibrated parameters for a Cream–Skin and Ointment–Skin in vitro permeation testing (IVPT) model. The table showcases the total number of parameters in the model that necessitated optimization. The model comprised approximately 16 distinct parameters, including the permeability and partition coefficients between various layers of the skin. Presently, the optimization procedure was only required for five of these parameters. Furthermore, the calibrated parameters showed minimal variation between the cream and ointment models, indicating that they can be considered as the definitive set of values for simulating Desoximetasone in the skin.

These findings suggest that the in silico models developed in this study are capable of accurately predicting the permeation behavior of Desoximetasone across the skin. The agreement between the simulated and experimental results provides further validation of the approach utilized in this study, and supports the potential of in silico modeling as a means of reducing the time and cost of drug development while improving the accuracy and reliability of predictions. However, it is worth noting that further research is necessary to evaluate the robustness and generalizability of the proposed approach across different drugs, formulations, and experimental conditions.

## 3. Materials and Methods

### 3.1. Materials

Desoximetasone standard was purchased from Ambeed, Inc. (Arlington Heights, IL, USA) Methanol, water (both HPLC grade), and acetic acid were purchased from Sigma Aldrich (St. Louis, MO, USA). Phosphate-buffered saline (PBS) was purchased from Gibco. The marketed Desoximetasone formulations used in this study were generics. Desoximetasone cream 0.25% was purchased from Perrigo (Dublin, Ireland) and Desoximetasone Ointment 0.25% was purchased from Actavis. For the permeation studies, dermatomed human cadaver skin was used, which was purchased from Skin Care, Inc. (Phoenix, AZ, USA).

### 3.2. Experimental Methods

#### 3.2.1. HPLC Method Development and Quantification

Desoximetasone was quantified using high-performance liquid chromatography (HPLC) using UV light [20]. The HPLC system included an Agilent 1100 Series liquid chromatography (Agilent Technologies, Santa Clara, CA, USA) and the Agilent Chemstation software (OpenLab CDS, ChemStation Edition, Rev. C.01.10, product version 5.0.0.352, Agilent Technologies). As a stationary phase, an Eclipse Plus C-18, 4.6 × 150 mm; 5 μm reverse-phase column was used. The column temperature was maintained at 30 °C. The mobile phase, methanol: water: acetic acid, mixed in the ratio of 65:35:1, was used in an isocratic method at a flow rate of 1 mL/min. The solution was degassed for 10 min. The UV detector was set at 254 nm for the detection of Desoximetasone. The linearity of Desoximetasone was checked from 0.02 µg/mL to 400 µg/mL with an R^2^ = 0.9998. The limit of detection (LOD) was 0.01 µg/mL and the Limit of Quantification (LOQ) was at 0.02 µg/mL. The inter and intra day variability was checked at room temperature and under refrigerated conditions.

#### 3.2.2. In Vitro Release Testing

Inert Snakeskin tubing membrane was purchased from Thermo Scientific Lot no. RF235434 (10 k MWCo, 16 mm dry I.D.). The membrane was stored at 4 °C and was thawed in Phosphate buffer saline (PBS, pH = 7.4) for approximately 30 min. The membrane was cut into 2 × 2 cm sections and transferred onto Franz Diffusion Cells (FDC) purchased from Logan Instruments, Somerset, NJ, USA. The receptor compartment of the FDC was filled with from 4.7 to 4.9 mL of PBS (pH 7.4) with a stirrer bar for a uniform distribution of temperature and media. The stirring was maintained at 700 rpm. The entire FDC setup was placed in a heat block (Logan Instruments, Somerset, NJ, USA) with the temperature maintained at 32 °C.

This entire setup was calibrated for 30 min under the conditions defined in the previous section before the application of the formulations [21,22]. The donor compartment of the FDC setup was dosed with 500 mg of Desoximetasone cream and ointment, and 500 mL of Desoximetasone solution 0.25% (*n* = 6 each). At selected time intervals, with respect to the time of dosing for each cell, an aliquot of receptor media (400 μL) and the same volume of fresh PBS were replaced in the receptor compartment. At each time interval, the receptor compartment was checked for air bubbles under the membrane. The concentrations of Desoximetasone were analyzed using the HPLC method (Section 2.2.1).

After 24 h, the membrane mounted on the receptor compartment was washed with methanol and collected in a vial with the remaining dose in the donor compartment. The entire volume was made to be 10 mL using methanol. The vials were then sonicated for 30 min and vortexed to release the Desoximetasone from the cream/ointment. The washed membrane exposed to the dose was cut out using scissors into small pieces and homogenized in 1 mL of methanol. The homogenized membrane was centrifuged at 10,000 rpm for 10 min. The supernatant was collected and filtered using 0.45 μm nylon filters. The supernatant was quantified for Desoximetasone using the HPLC method discussed in Section 2.2.1.

#### 3.2.3. Ex Vivo Skin Permeation Studies

Dermatomed full-thickness human cadaver skin purchased from Skin Care (Pheonix, AZ, USA) was used for the permeation studies. The skin was cryopreserved at −80 °C upon receipt. The skin was thawed at room temperature in PBS (pH = 7.4) for approximately 30 min. The skin was sectioned into approximately 2 × 2 cm^2^ pieces, which were mounted onto Franz Diffusion Cells (FDC) with the Stratum Corneum (SC) facing the donor chamber (Figure 8B). The receptor compartment was filled with 4.9 mL of PBS (pH = 7.4) with a stirrer bar. The stirring was maintained at 700 rpm. The temperature of the dry heat block (Logan Instruments, Somerset, NJ, USA) was maintained at 37 °C and the temperature of the skin was maintained at 32 °C (Figure 8C).

The system was equilibrated for 30 min before the application of the dose [23,24,25]. Post-equilibration, the FDC set up was dosed with 500 mg of Desoximetasone cream and ointment (*n* = 5 each). At selected time intervals, with respect to the time of dosing for each cell, an aliquot of receptor media (400 μL) and the same volume of fresh PBS were replaced in the receptor compartment. At each time interval, the receptor compartment was checked for air bubbles under the skin. The concentrations of Desoximetasone were analyzed using the HPLC method (Section 2.2.1).

After 24 h, the skin mounted on the receptor compartment was washed with methanol and collected in a vial with the remaining dose in the donor compartment. The entire volume was made to be 10 mL using methanol. The vials were then sonicated for 30 min and vortexed to release the Desoximetasone from the cream/ointment. The washed skin exposed to the dose was cut out using scissors and the dermis and epidermis layers were separated using forceps (Figure 8D). The dermis and epidermis were cut into small pieces and homogenized in 1 mL of methanol. The homogenized skin was centrifuged at 10,000 rpm for 10 min. The supernatant was collected and filtered using 0.45 μm nylon filters. The supernatant was quantified for Desoximetasone using the HPLC method discussed in Section 2.2.1. (Figure 8E).

To study the equivalence and recovery of the applied Desoximetasone, the amount of Desoximteasone from each compartment in the FDC was added.
Total Desoximetasone calculated=Residual formulation in donor compartment+Receptor cell media+Desoximetasone in dermis+Desoximetasone in Epidermis.
Percentage of Desoximetasone recovered=Total Desoximetasone calculated/Desoximetasone in formulation

#### 3.2.4. Statistical Analysis

The data are reported as mean ± standard deviation (S.D.) (*n* = 5). The obtained results were analyzed using a One-Way Analysis of Variance (ANOVA). The statistical significance was based on a 95% confidence interval.

### 3.3. Computational Methods

All the computational models were developed using CFD Research’s Computational Biology (CoBi) framework, which is a Department of Defense (DoD) open-source C++ platform for multiscale multiphysics modeling. CoBi has previously been used for various physiologically based pharmacokinetic (PBPK) applications [3,26,27].

#### 3.3.1. Virtual Skin Model

The skin model developed by CFD Research is an integrated dermal absorption model that combines a multi-compartmental model of skin with a whole-body PBPK model to predict the permeation and clinical pharmacokinetics of transdermal drug delivery systems. The model is holistic in nature and accounts for all the different layers of the skin, including the stratum corneum, epidermis, and dermis. The stratum corneum is described in high resolution, with lipids and corneocytes being explicitly modeled using a brick-and-mortar approach. The epidermis and dermis layers are discretized for accurate transport in these layers. The model also includes first-principles-based mechanistic models for different dermal drug delivery systems, which are coupled with the dermal model to investigate the release and permeation behavior in the skin for different dosage forms. Additionally, the integrated dermal model is linked to a systems pharmacology model to predict the dermal absorption and clinical pharmacokinetics of dermally delivered compounds. These stand-alone and integrated whole-body-dermal models are validated against in vitro and in vivo experimental data from the literature. Further information on the stand-alone dermal model, whole-body-dermal model, and corresponding model validation against the literature data can be found in the authors’ previously published manuscript [3]. Figure 9 summarizes the dermal anatomy and different model capabilities.

#### 3.3.2. IVRT Computational Model Setup

In this study, the in vitro release testing (IVRT) experimental setup was computationally replicated using the CoBi Platform. The different domains are represented using compartments. The drug concentrations in the different compartments were governed through interface fluxes modeled using diffusive transport (Equation (1) in Table 3). A mathematical model that represented the volume sampling and replenishing was integrated into the receptor compartment model to accurately replicate the concentration dynamics in the receptor compartment. The different equations used in the model development are summarized in the table below.

During the model setup, experimental data corresponding to Desoximetasone Cream 0.25% (Actavis) and Desoximetasone Ointment 0.25% (Perrigo) were utilized for the development and optimization of the corresponding in silico release model. Some of the release model parameters were calibrated by comparing them against the experimental data.

Separate computational models were developed to represent both the cream and ointment. Creams are generally defined as semi-solid emulsions, with oil droplets dispersed in a water continuous phase. This is because creams typically contain a mixture of hydrophilic and lipophilic components, which require the use of an emulsifying agent to create a stable mixture. As a result, modeling creams as O/W emulsions is a logical and widely accepted approach. Ointments are typically composed of a viscous base, such as petrolatum, that serves as a continuous phase for dispersed solid particles. Ointments do not require an emulsifying agent, as the particles are stabilized by the viscosity of the base. Desoximetasone, the active ingredient in the ointment, exists in a solid form and is therefore more likely to be dispersed as solid particles rather than as liquid droplets.

Therefore, a two-phase diffusion model, similar to the model developed by Bernardo et al. [28], was employed to simulate the drug release from the cream. The ointment was assumed to be a type of suspension. The drug release from the ointment was simulated using the Nernst Brunner Modification of the Noyes–Whitney model. Table 4 shows the corresponding model information. The diffusivity across the membrane was estimated using the Mackie–Meares formula, which accommodates the porosity of the membrane. Further research in this area could expand upon the current study by incorporating additional factors that may impact drug release and permeation, such as the effect of the formulation parameters and the impact of co-solvents.

#### 3.3.3. IVPT Computational Model Setup

In this study, the validated and optimized release model parameters were implemented in the in silico modeling tools of CoBi to replicate the in vitro permeation testing (IVPT) experiments. The computational model was set up based on the available information on the experimentally used Franz Cell setup, as shown in the schematic below in Figure 10. The skin thickness was adjusted to match the average specimen thickness used in the experiments by reducing the dermis thickness. To ensure the accurate replication of the concentration dynamics in the receptor compartment, a mathematical model representing the volume sampling and replenishing was integrated into the model. The different governing equations simulating the transport in the skin model can be obtained from our prior work [3] and the different input model parameters are listed in Table 5.

## 4. Discussion

In this study, the authors proposed an innovative methodology for optimizing the parameters of a mechanistic virtual dermal absorption model by utilizing the release and permeation data of Desoximetasone in different formulations. The experiments conducted provided new insights into the release kinetics and permeation profiles of Desoximetasone in different dosage forms. The findings from these experiments were then used to develop a computational model for optimization purposes. The proposed virtual in vitro permeation testing (IVPT) model was successfully validated, and can be used to predict drug release and permeation during the drug development process.

The key findings of this study demonstrated that there were differences in the release profiles of Desoximetasone from cream (Actavis) and ointment (Perrigo) dosage forms, as well as differences in the permeation profiles in the skin. Specifically, the IVRT experiments showed that the cream formulation released a higher amount of Desoximetasone compared to the ointment formulation, which was consistent with previous knowledge about the different properties of these formulations. In addition, the IVPT experiments revealed the differences in the permeation profiles of Desoximetasone through the skin, indicating that the properties of formulations can affect their ability to penetrate the skin. These findings highlight the importance of considering the physicochemical properties of drug formulations in the development and optimization of topical products.

The key findings demonstrated some key differences in the release and permeation profiles. The release profiles indicated that the Desoximetasone from the cream permeated through the membrane faster than the ointment. However, the difference in the release of the two formulations was not significant. However, the cumulative amount of Desoximetasone at the end of the 24 h study was also significantly higher in the cream with respect to the ointment. Additionally, it is noteworthy that the release of Desoximetasone from methanol was significantly lower as compared to both the cream and ointment, indicating the role of excipients in formulations.

Additionally, when these results were compared with the permeation study conducted on the human cadaver skin, the cumulative amount of Desoximetasone in the receptor media was comparable at 36 h. However, the statistical analysis of flux at each time point indicated that the permeation of Desoximetasone from the cream was significantly higher from 14 h to 18 h. However, before and after these time points, no significant difference was seen in two the formulations. From this finding, it is noteworthy to note differences in the lag time of Desoximetasone from a water-based formulation (cream) and oil-based formulation (ointment). This effect may be attributed to the hydrophilic pull from the hydrophilic dermis in the case of the cream when compared to an ointment. This effect of the difference in the hydration concentrations of the skin layers and the interplay with the formulations may also explain the delay in the permeation of Desoximetasone through skin, as indicated in Figure 3. It is also noteworthy that Desoximetasone was detected in the receptor media much later with the application of ointment as compared to cream. This can also be attributed to the basic chemistry of the formulations, where the water concentration in cream is higher as compared to ointment. The higher water concentration may have driven the better partitioning of Desoximetasone (logP of 2.3) from a relatively hydrophilic environment to the lipophilic epidermis. Additionally, water is also considered to be a permeation enhancer.

The amount of Desoximetasone deposited in the skin layers was also studied post ex vivo permeation studies. The amount of Desoximetasone calculated per weight of the skin indicated that Desoximetasone tends to deposit in the epidermis as compared to the dermis. No significant difference was seen in deposition when the cream and ointment formulations were compared. This can be attributed to the lipophilic logP (2.3) of Desoximetasone, which will have the tendency to interact with more lipophilic epidermis than hydrophilic dermis.

The current study introduces virtual models for in vitro drug release and permeation testing, utilizing a mechanistic approach based on first principles. This approach provides a better understanding of the underlying physicochemical mechanisms involved in drug release and permeation, leading to an improved accuracy in predicting these processes. However, a significant challenge in using a mechanistic approach is the extensive requirement for experimental data to calibrate the model parameters. Specifically, as the complexity of the model increases, the amount of necessary experimental data grows, and a lack of appropriate experimental data limits the ability to calibrate these models effectively. The authors acknowledge that model optimization is a considerable issue due to the need for comprehensive experimental data.

The authors developed a more systematic optimization approach for virtual IVPT models that involves using both IVRT and IVPT data. This method optimizes the model parameters in two stages, starting with the optimization of the dosage model, followed by the optimization of the skin model. In the first stage, the authors utilized IVRT experimental data to calibrate the dosage model by adjusting various model parameters, such as the diffusivity, partition coefficient, droplet size, and others. The authors’ use of both IVRT and IVPT data in the optimization process provided a more comprehensive understanding of the drug release and permeation profiles. This approach addresses a significant gap in previous studies, which have often focused solely on using single set of experimental data for model optimization. The authors’ use of a two-stage optimization process, starting with the dosage model, ensured that the model was calibrated accurately and effectively, providing more reliable predictions of the drug release and permeation profiles. This approach has significant implications for the development and optimization of new drug formulations, where accurate predictions of drug release and permeation profiles are crucial for successful clinical outcomes.

The authors developed two different models to represent the cream and ointment formulations, respectively, including (a) a two-phase (oil droplets in water) diffusion-type model for simulating the drug release from the cream and (b) a solid particle (Nersnt–Brunner modification of the Noyes–Whitney release model) in oil-base-type model for the ointment model. Generally, creams are a type of emulsion that typically contain oil and water as their two main components, with an emulsifying agent added to help keep the mixture stable. Ointments, on the other hand, are typically not emulsions, but rather a mixture of an oil or a petrolatum base with the active ingredient. Desoximetasone is a crystalline powder at room temperature, but it is dissolved in the cream base to form a homogeneous mixture. The cream is a uniform and stable mixture in which the Desoximetasone is dissolved in the cream base, rather than in solid form. In contrast, during the preparation of the ointment, the Desoximetasone powder is dispersed and uniformly distributed throughout the molten ointment base. Desoximetasone is generally considered to be poorly soluble in water and organic solvents. Based on this information, it is safe to assume that the majority of the Desoximetasone is in solid form and uniformly distributed throughout the ointment base while it is in the form of oil droplets dispersed in the cream (water) base.

Upon the model optimization, it was observed that the optimized diffusivity in the ointment (1 × 10^−10^ m^2^/s) was slightly greater than that of the cream (0.83 × 10^−10^ m^2^/s). In general, the diffusivity of a solute in a cream is expected to be higher than in an ointment due to the difference in their viscosity and particle size distribution. Creams are typically less viscous and have smaller particle sizes, which can facilitate the diffusion of the solute through the medium. While a higher diffusivity for ointment compared to cream is unexpected, it is possible that the specific properties of the formulations or solutes being used could explain this result. Further experimentation and analysis may be needed to fully understand the differences in diffusivity observed between the two formulations.

Following the optimization of the dosage model, the calibrated parameters were employed in the virtual in vitro permeation testing (IVPT) model to simulate the permeation of Desoximetasone. The study determined that the model accurately predicted the permeation data with minimal optimization requirements, highlighting the efficacy of the approach utilized. The authors’ method represents a significant advancement towards establishing more reliable and accurate virtual in vitro release testing (IVRT) and IVPT models. Despite potential challenges in acquiring essential experimental data, the systematic approach taken for model optimization in this study can minimize the complexity of optimization and enhance the precision and dependability of virtual models.

While the computational model presented in this study is a significant step towards understanding the mechanisms of drug release from cream and ointment formulations, there are certain limitations that should be acknowledged. Firstly, the model did not explicitly account for the viscosity of the formulations, which could impact the drug release and penetration through the skin. This could be addressed by incorporating viscosity into the model, although this may require additional experimental data to establish the appropriate parameters. Additionally, while the model is capable of handling multiple particle sizes, this information was not included in the simulations due to a lack of available experimental data. Further research is needed to establish how particle size affects the drug release and penetration of creams and ointments, and this information could be incorporated into future versions of the model. Another limitation is the lack of information on the composition of the formulations, which was not accounted for in the model. If detailed information on the composition of the formulations is available, this could enable the calculation of diffusivities and other key parameters using a more mechanistic approach. This could potentially enhance the accuracy and reliability of the model.

Future work in this area can build upon the current study by investigating several avenues for the further development of the computational model. One important direction for future work would be to gather additional experimental data to more accurately calibrate the model parameters [16,17,29]. This could involve obtaining data on the viscosity, particle size distribution, and composition of the formulations, as well as the diffusion coefficients and partition coefficients of the solute. Such data would help to enhance the accuracy and reliability of the model predictions. Another potential area for future work would be to investigate the impact of the co-solvent effects on the drug release from cream and ointment formulations. Co-solvents are commonly used in topical formulations to enhance solubility and drug delivery, and their effects on drug release and permeation could be incorporated into the computational model. Additionally, the CoBi-Q3D model could be utilized to simulate the release of multiple solutes, which would enable more complex and realistic simulations of drug release from topical formulations. Other avenues for future work could include exploring the effects of the formulation parameters, such as emulsifying agents, surfactants, and stabilizers, on drug release. The model could also be extended to simulate the drug release from other topical formulations, such as gels, foams, and lotions. Another potential avenue for future work is the incorporation of experimental data from other drug delivery systems, such as iontophoresis or microneedle-mediated drug delivery [30,31,32]. These data could be used to further improve the computational model and extend its applicability to different drug delivery systems. By expanding the scope of the model, it could become a more versatile tool for predicting drug release and optimizing the drug delivery for a wider range of drug products.

## 5. Conclusions

This study presented an innovative methodology for optimizing the parameters of a mechanistic virtual dermal absorption model using the release and permeation data of Desoximetasone in different formulations. The experiments conducted provided important insights into the release kinetics and permeation profiles of Desoximetasone, which were then used to develop a computational model for optimization purposes. The proposed virtual in vitro permeation testing (IVPT) model was successfully validated and has the potential to improve the accuracy and efficiency of drug development processes by predicting drug release and permeation. In addition, the findings of this study can be applied to other drugs and formulations, while further research could incorporate additional factors that may impact drug release and permeation.

## Figures and Tables

**Figure 1 ijms-24-15118-f001:**
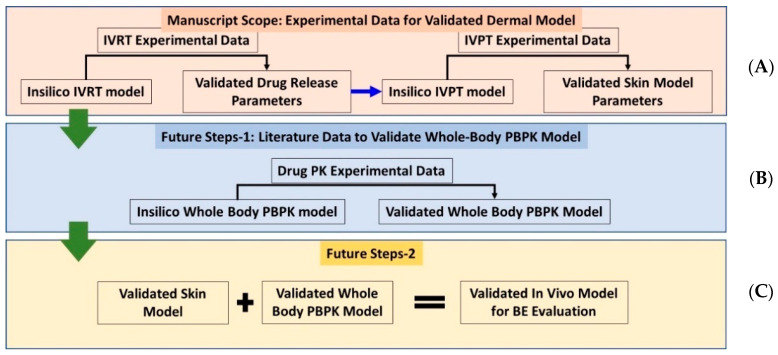
Schematic showing (**A**) Current approach, discussed in this manuscript, to validate the stand-alone in vitro dermal modeling components and (**B**,**C**) future vision for development of validated whole-body-dermal PBPK models for virtual bioequivalence (BE) evaluation.

**Figure 2 ijms-24-15118-f002:**
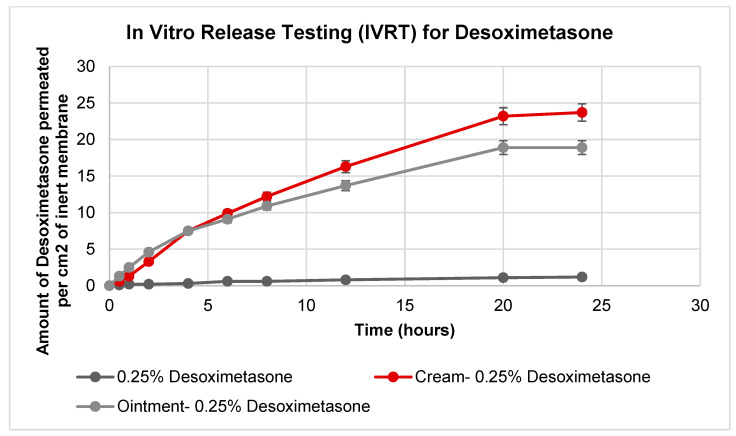
Amount of Desoximetasone permeated per cm^2^ of inert membrane. In total, 0.25% of Desoximetasone was prepared by diluting stock (prepared in methanol) in water. (*n* = 6).

**Figure 3 ijms-24-15118-f003:**
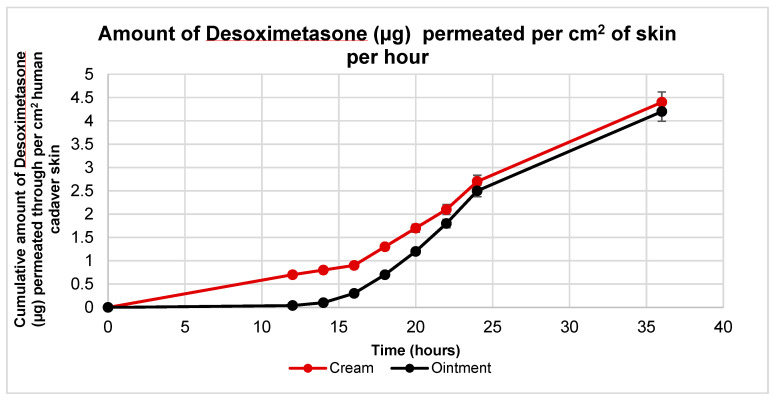
Ex vivo permeation profile of Desoximetasone per cm^2^ of human cadaver skin from Rx cream and ointment in a 36 h permeation study. The area for Desoximetasone permeation was 0.64 cm^2^. Data plotted are represented as mean standard deviation (*n* = 5).

**Figure 4 ijms-24-15118-f004:**
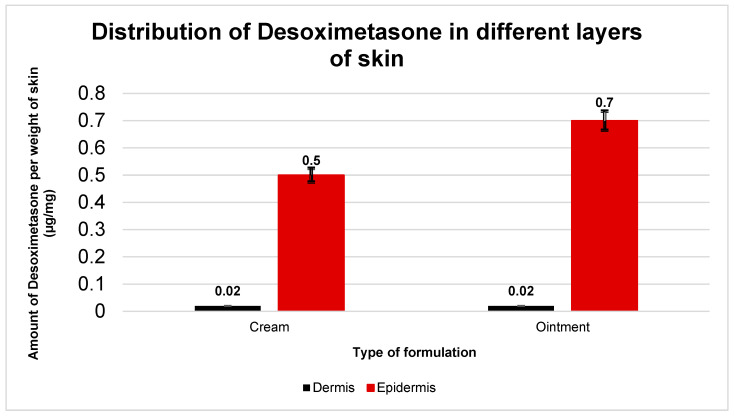
Amount of Desoximetasone detected in epidermis and dermis of human cadaver skin after 36 h permeation study (*n* = 5). The graphs are represented as amount of Desoximetasone (μg) per weight of skin layer (mg). Data plotted is as mean standard deviation. Desoximetasone deposition in Cream: epidermis = (0.5 ± 0.1) µg/mg; and dermis = (0.02 ± 0.003) µg/mg. Ointment: epidermis = (0.7 ± 0.2) µg/mg; and dermis = (0.02 ± 0.003) µg/mg.

**Figure 5 ijms-24-15118-f005:**
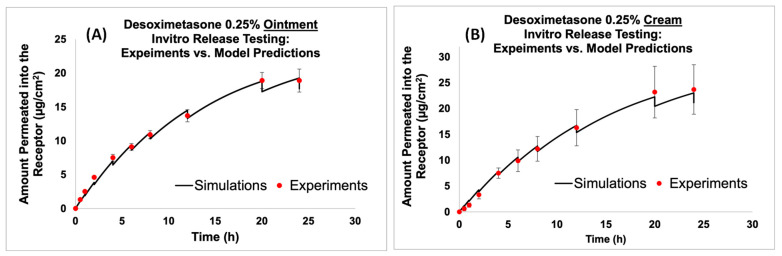
Virtual IVRT model predictions compared against experimental data for ointment and cream formulations. (**A**) Comparison of the model prediction of amount permeated into the receptor to experimental data for Desoximetasone Ointment (0.25%), and (**B**) comparison of the model prediction of amount permeated into the receptor to experimental data for Desoximetasone Cream (0.25%).

**Figure 6 ijms-24-15118-f006:**
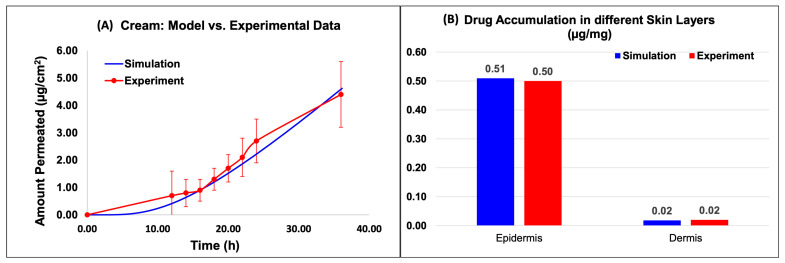
Virtual IVPT model predictions compared against experimental data for the cream formulation. (**A**) Comparison of the model prediction of amount permeated into the receptor to experimental data, and (**B**) comparison of the API accumulation in the different skin layers.

**Figure 7 ijms-24-15118-f007:**
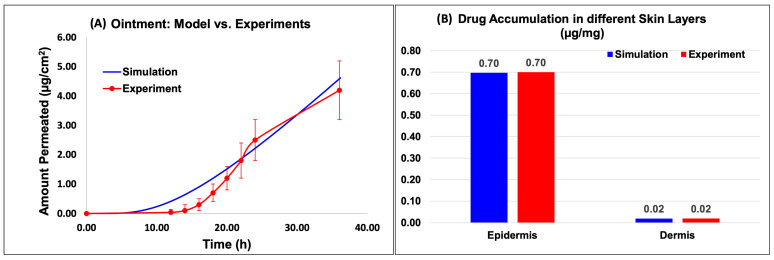
Virtual IVPT model predictions compared against experimental data for the ointment formulation. (**A**) Comparison of the model prediction of amount permeated into the receptor to experimental data, and (**B**) comparison of the API accumulation in the different skin layers.

**Figure 8 ijms-24-15118-f008:**
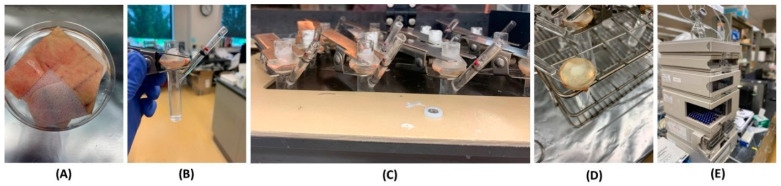
(**A**) Cryopreserved dermatomed human cadaver skin of 500 µm thickness stored at −80 °C is thawed at room temperature using PBS (pH = 7.4), (**B**) the skin is section into approximately 2 × 2 cm pieces. The sectioned skin is mounted onto Franz diffusion cell (FDC) and clamped. The donor is loaded with cream (*n* = 5), ointment (*n* = 5), or control solution (*n* = 2), respectively, (**C**) the FDC is transferred into the incubator. Skin surface temperature is maintained at 32 °C, (**D**) after 36 h, the clamps are removed, and the residual dose is collected from the donor chamber. The skin is washed with HPLC-grade methanol to collect residual dose. Skin layers are separated (dermis and epidermis), and (**E**) quantification of drug permeation using a validated HPLC method.

**Figure 9 ijms-24-15118-f009:**
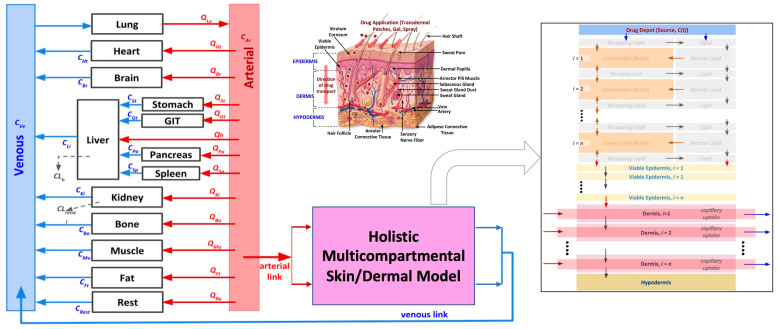
Integrated whole-body-holistic-skin PBPK model developed by CFD Research [3]. The different arrows in this figure represent the flow and diffusive transport between different compartments. The red and blue arrows (in the whole-body representation) show the flow between the different compartments. The arrows in the virtual skin model represent the transport between the different skin layers.

**Figure 10 ijms-24-15118-f010:**
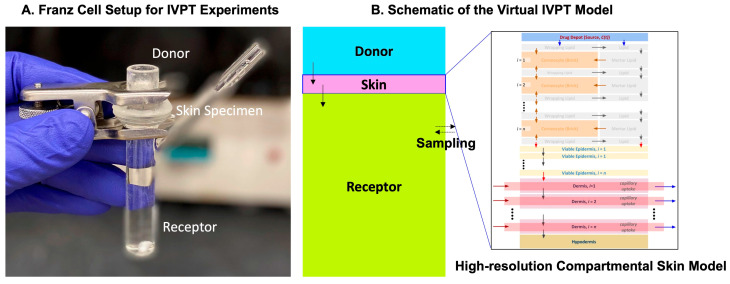
(**A**) Franz Cell setup for the IVPT experiments conducted by the Rutgers team and the corresponding virtual IVPT model schematic developed by the CFDRC team. (**B**) A high-resolution skin model was used to describe the skin specimen for the virtual IVPT model.

**Table 1 ijms-24-15118-t001:** The flux of Desoximetasone in cream is significantly higher from 14 h to 18 h after the application of formulation on human cadaver skin.

Time (Hours)	*p*-Value	F-Stat
0	0	0
12	0.1479	2.5657
14	0.0231	7.8625
16	0.0149	9.5416
18	0.0438	5.7154
20	0.146	2.5929
22	0.407	0.7658
24	0.7935	0.0733
36	0.803	0.0665

Similar trend was seen in permeation of Desoximetasone from cream and ointment. The amount of Desoximetasone permeated at the end of 36 h IVPT was comparable in cream and ointment.

**Table 2 ijms-24-15118-t002:** Table shows the different optimized model parameters for virtual IVPT model with Cream and Ointment formulations of Desoximetasone. All units are in S.I. system.

Model Parameter	Mechanistic Estimation	Optimized Cream–Skin	Optimized Ointment–Skin
K_p_ (Lipid/Vehicle)	18.8	18.8	11.6
Permeability (Lipid/Corneocyte) Horizontal (m/s)	4.15 × 10^−11^	1.04 × 10^−10^	0.87 × 10^−10^
Permeability (Lipid/Corneocyte) Vertical (m/s)	2.07 × 10^−9^	5.2 × 10^−9^	4.35 × 10^−9^
Viable Epidermis Diffusivity (m^2^/s)	3.65 × 10^−11^	1.1 × 10^−9^	1.1 × 10^−9^
Dermis Diffusivity (m^2^/s)	3.65 × 10^−11^	1.1 × 10^−9^	1.1 × 10^−9^
Permeability (Dermis/Receptor) (m/s)	7.02 × 10^−9^	3.5 × 10^−7^	3.5 × 10^−7^

**Table 3 ijms-24-15118-t003:** Table summarizing the governing equations used in developing an in silico model of the in vitro release testing setup.

Equation No.	Equation	Explanation
(1)	$\begin{array}{c} J_{A\to B}=P_{A\to B}.SA_{A/B}\left(C_{A}-\frac{C_{B}}{K_{p,A/B}}\right)\\ P_{A\to B}=\frac{1}{\frac{\delta_{A}}{K_{p,A/B}D_{A}}+\frac{\delta_{B}}{D_{B}}} \end{array}$	Generalized diffusive transport equation used in the model to simulate the transport between two compartments, A and B. In this equation, *J* is the diffusion flux between compartments *A* and *B* *P* is the permeability *D* is the diffusivity *δ* is the diffusion distance *K_p_* is the partition coefficient (*A*/*B*) *SA* is the interface surface area *C* is the drug concentration This equation is used to model the transport between different compartments, including oil droplets or suspension particles, vehicle, skin, receptors, and others. The drug release from the oil droplets was simulated using the same governing equations (with corresponding drop volume).
(2)	$J_{P\to V}=N_{p}D_{P}SA_{P}\left(\frac{C_{S}-C_{V}}{R_{P}}\right)$	Drug release from suspension particles (ointment) using the Nernst Brunner modification of the Noyes–Whitney equation. In this equation, *J* represents the flux from the particle to the vehicle *N* is the particle number *D* is the diffusivity at the surface of the particle *SA* is the particle surface area *C_s_* is the solubility *C_v_* is the vehicle drug concentration *R_p_* is the particle radius.
(3)	$V_{R}\frac{dC_{R}}{dt}=J_{input}-(Q_{Sampling}C_{R})$	Equation used to simulate the sampling effects in receptor compartment. In this equation, *Q_Sampling_* is the sample volume removal from the receptor compartment. The sampling process affects the overall concentration of the receptor. *J_input_* is the total effective input diffusive flux into the receptor.

**Table 4 ijms-24-15118-t004:** Input parameters for in silico IVRT models for ointment and cream.

Model Component	Parameters	Value
Desoximethasone	Molecular Weight	376.5 g/mol
	LogP	2.35
	pH	7.4
	pKa	13.44
Cream/Ointment	Thickness	0.92 cm
	Initial Mass	1250 µg
Ointment (solid dispersed in liquid)	Diffusivity (In Vehicle)	1 × 10^−10^ m^2^/s
	Particle Solubility	0.63
	Particle Size	~3.37 µm
	Partitioning (Vehicle/Receptor)	223.872
Cream (liquid in liquid)	Diffusivity (In Vehicle)	0.85 × 10^−10^ m^2^/s
	Partitioning (Continuous Phase/Dispersed Phase)	4.484
	Droplet Size	~3.37 µm
	Partitioning (Vehicle/Receptor)	223.872

**Table 5 ijms-24-15118-t005:** Input model parameters for the virtual IVPT experiments.

Model Component	Parameters	Value
Desoximethasone	Molecular weight	376.5
	logP	2.35
	pH	7.4
	pKa	13.44
Vehicle (Cream/Ointment)	Thickness	0.92 cm
	Initial mass	1250 µg
Stratum Corneum	Thickness	14.075 µm
	Corneocyte diameter	40 µm
	Corneocyte thickness	0.8 µm
	Lipid thickness	0.075 µm
Viable Epidermis	Thickness	~56 µm
Dermis	Thickness	~1.2 mm
Receptor	Volume	4.7 mL
Receptor	Sampling point	Center of the receptor compartment

## Data Availability

No new data were created or analyzed in this study. Data sharing is not applicable to this article.

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
