# Peer review of "A Comparative Evaluation of Desoximetasone Cream and Ointment Formulations Using Experiments and In Silico Modeling"

_ijms, 2023, doi:10.3390/ijms242015118_

Round 1
Reviewer 1 Report
The manuscript is poorly written and lacks references in the introduction and discussion. It introduces the topic of new research but needs to improve in several areas; these issues must be addressed to make it interesting for the journal readers. The author should justify some points.
Point 1: There is a need to clarify the central question addressed by the research.
Point 2: Authors must address the field's specific research gap filled by research.
Point 3: Add to the subject area compared with other published material.
Point 4: Abstract is not sufficient of the present work; it should be elaborate with present findings.
Point 5: All abbreviations should expand in the manuscript.
Point 6: There is a need to use PBBM modelling with PBPK for a range of biopharmaceutical applications. Why did the authors not use the PBBM modelling? clarify it.
Point 7: There is a need to add the reference to the first paragraph.
Point 8: Line101: Explain “ Error! Reference source not found”.
Point 9: There is a need to rewrite the last paragraphs of the introduction with the addition of findings and limitations of the study.
Point 10: There is a need to re-examine the Table 3.
Point 11: There is a requirement to do the Equivalence test. Why did it not conduct?
Point 12: Discuss the application/theory/method/study reported in sufficient detail to allow for its replicability and reproducibility.
Author Response
Dear Reviewer,
Thank you for the comments and you time.
We have tried to address all the comments. Kinda find them attached. Please let us know if any further revisions are needed.

Reviewer 2 Report
The authors investigated Desoximetasone-containing cream and ointment in the manuscript with a new virtual dermal absorption model that predicts drug release and permeability data. The manuscript is logically compiled, and the tables and figures are informative.
Some critical comments:
- In many cases, the literary references in the article should be clarified (e.g. there are such errors on pages 3; 6; 7; 11; 12; and 14).
- None of the parameters shown in Table 4 are units of measure included. As far as I know, diffusivity is a parameter with units.
- Check the order of the numbering of the tables.
- The number of parallel measurements is not included in the method descriptions.
- In my opinion, a list of abbreviations would be useful.
- The SD values are missing from Figure 5, and it would be advisable to replace the word polymer with the term inert membrane in the figure title. - In the case of the simulation connection, the line is not lucky, because the amount of the active ingredient seems to decrease at a given time.
In vitro correctly, not invitro
There is u in several places (page 4; 138 and 173 lines), but it is just a prefix.
Author Response
Dear Reviewer,
Thank you for your comments and time.
We have tried to address all the comments. Kindly find them attached.

Round 2
Reviewer 1 Report
Most of the suggestions are incorporated in the revised version of the manuscript. It could be published.
Author Response
Hello,
Please find the revised manuscript attached.
Best regards,
Namrata
